

# Rapid relative increase of crustose coralline algae following herbivore exclusion in a reef of El Salvador

Xochitl E. Elías Ilosvay[1], Johanna Segovia[2], Sebastian Ferse[1,3], Walter Ernesto Elias[4] and Christian Wild[1]

[1] Faculty of Biology and Chemistry, Marine Ecology Department, University of Bremen, Bremen, Deutschland
[2] Universidad Francisco Gavidia, San Salvador, El Salvador
[3] Department of Ecology, Leibniz Centre for Tropical Marine Research (ZMT), Bremen, Germany
[4] Unaffiliated, Antiguo Cuscatlán, La Libertad, El Salvador

## ABSTRACT

The Eastern Tropical Pacific (ETP) is one of the most isolated and least studied regions in the world. This particularly applies to the coast of El Salvador, where the only reef between Guatemala and Nicaragua, called Los Cóbanos reef, is located. There is very little published information about the reef's biodiversity, and to our knowledge, no research on its ecology and responses to anthropogenic impacts, such as overfishing, has been conducted. The present study, therefore, described the benthic community of Los Cóbanos reef, El Salvador, using the Line-Point-Intercept-Transect method and investigated changes in the benthic community following the exclusion of piscine macroherbivores over a period of seven weeks. Results showed high benthic algae cover (up to 98%), dominated by turf and green algae, and low coral cover (0–4%). *Porites lobata* was the only hermatypic coral species found during the surveys. Surprisingly, crustose coralline algae (CCA) showed a remarkable total cover increase by 58%, while turf algae cover decreased by 82%, in experimental plots after seven weeks of piscine macroherbivore exclusion. These findings apparently contradict the results of most previous similar studies. While it was not possible to ascertain the exact mechanisms leading to these drastic community changes, the most likely explanation is grazing on turf by small grazing macroherbivores that had access to the cages during the experiment and clearing of CCA initially covered by epiphytes and sediments. A higher CCA cover would promote the succesful settlement by corals and prevent further erosion of the reef framework. Therefore it is crucial to better understand algal dynamics, herbivory, and implications of overfishing at Los Cóbanos to avoid further reef deterioration. This could be achieved through video surveys of the fish community, night-time observations of the macroinvertebrate community, exclusion experiments that also keep out herbivorous macroinvertebrates, and/or experimental assessments of turf algae/CCA interactions.

Corresponding author
Xochitl E. Elías Ilosvay,
xochitl.ilosvay@gmail.com

## INTRODUCTION

Many tropical reefs around the world are undergoing changes in benthic community composition (away from dominance by stony corals) as a result of the combined effect of anthropogenic disturbances (*Hoegh-Guldberg et al., 2018*; *Hughes et al., 2007*; *Pandolfi et al., 2005*). Studies show that by removing herbivores that consume macroalgae, overfishing may favor competitive fleshy algae and turf over corals and other reef-building organisms (*Burkepile & Hay, 2009*; *Hughes et al., 2007*; *Smith, Hunter & Smith, 2010*). Reef herbivores directly affect the composition of reef benthic communities by freeing benthic space from macroalgae and allowing, for example, coral larval settlement (*Lewis, 1986*; *Steneck, 1995*) and crustose coralline algae (CCA) growth (*Mumby, 2009*). The latter plays a crucial role in coral reef ecosystems by facilitating the settlement of coral larvae via chemical cues (*Heyward & Negri, 1999*; *Ritson-Williams et al., 2010*), solidifying the reef framework (*Adey, 1998*; *Weiss & Martindale, 2017*), and preventing bioerosion (*Weiss & Martindale, 2017*).

Several studies address the role that herbivorous fish and invertebrates play in the coral reefs of the Caribbean Sea and the Indo-Pacific (e.g., *Foster, 1987*; *Green & Bellwood, 2009*; *Hughes et al., 2007*; *Lewis, 1986*). The Caribbean Sea, for example, suffered a drastic decrease in coral cover after mass mortality of the sea urchin, *Diadema antillarum*, preceded by overfishing of herbivorous fishes on many Caribbean reefs (*Jackson et al., 2014*), while the Indo-Pacific is characterized by a large number of diverse herbivorous fishes, showing higher functional diversity among herbivorous fish than the Caribbean (*Roff & Mumby, 2012*).

On the contrary, little research has been conducted on coral reefs of the Eastern Tropical Pacific (ETP), which is one of the most isolated ocean regions in the world (*Cortés et al., 2017*; *Glynn & Ault, 2000*). The reefs in the region are exposed to extreme environmental conditions, such as high $CO_2$ concentrations, low aragonite saturation, high levels of nutrients, high tidal amplitudes, and extreme fluctuations in seawater temperature caused by the El Niño-Southern Oscillation (*Bennett, 1966*; *Cortés, 1997*; *Guzmán & Cortés, 1993*; *Kessler, 2006*). Moreover, there is little research on how herbivores structure benthic communities in the ETP. Only a few studies have investigated the role of consumers through exclusion experiments in the region (e.g., *Menge, Lubchenco & Ashkenas, 1986*; *Vinueza et al., 2006*; *Vinueza et al., 2014*; *Roth et al., 2015*).

Los Cóbanos reef, located in El Salvador, is the only known reef in the country with hermatypic coral species. Los Cóbanos lies within the so-called "Pacific Central American Faunal Gap (PCAFG)", which is the coastal stretch between Guatemala and northwestern Nicaragua (*Cortés et al., 2017*). Los Cóbanos reef, together with the recently discovered reef at the coast of Nicaragua, are the only two sites within the PCAFG that have been found to host significant coral communities (*Alvarado et al., 2010*; *Reyes-Bonilla & Barraza, 2003*).

There is little information about Los Cóbanos reef. *Reyes-Bonilla & Barraza (2003)* reported eight reef-building coral species belonging to the genera *Porites*, *Pocillopora*, and *Pavona*. Unpublished management reports and surveys document that the reef is dominated by algae (~77%) (*Segovia & Navarrete Calero, 2007*), with 81 algae species

reported (*Arrivillaga et al., 2010*), and that hard coral cover is extremely low (∼4%) (*Segovia, 2016*). According to *Reyes-Bonilla & Barraza (2003)*, the fish abundance on the reef is low, but generally, information about the fish community at Los Cóbanos is scarce. Fish play a crucial role in coral reef systems by, for example, controlling macroalgae which compete with corals for space (*Bellwood et al., 2004*; *Mumby, 2016*). Overfishing in coral reefs can increase overgrowth of algae and/or other benthic organisms and lead to phase shifts from coral dominance to degraded ecosystems (*Bellwood et al., 2004*; *Loh et al., 2015*; *Pandolfi et al., 2005*). Currently, the literature states that around 137,000 kg of fish are being caught at Los Cóbanos (*Arrivillaga et al., 2010*). Although several studies (e.g., *Molina and Vásquez-Jandres, 2006*; *Segovia & Navarrete Calero, 2007*) state that Los Cóbanos reef is overfished, to our knowledge, no research has been conducted investigating how the piscine community affects benthic community composition at Los Cóbanos reef. This study aimed to address those knowledge gaps by, firstly, describing the current benthic community state of Los Cóbanos reef, and secondly, assessing the effect of piscine macroherbivore exclusion on the benthic community using an in-situ exclusion experiment, simulating overfishing. We hypothesized that the reef would exhibit high algae and low coral cover (H1), and that by excluding piscine macroherbivores, macroalgae would overgrow *P. lobata* colonies as suggested by previous studies (*Hughes et al., 2007*; *Roth et al., 2015*; *Thacker, Ginsburg & Paul, 2001*) (H2).

## MATERIALS & METHODS

### Study site

The study was carried out on Los Cóbanos reef (13°31′25.6″N 89°48′24.6″W), El Salvador, from March to May 2018. The reef lies within the nature reserve "Complejo Los Cóbanos", 11 km east of the city Acajutla (Fig. 1). The Ministry of Environment and Natural Resources of El Salvador (MARN, initials in Spanish) approved the fieldwork for this study inside the nature reserve. Los Cóbanos reef consists of a heterogeneous basalt shore with a tidal variation of ∼3 m (*Segovia, 2016*). Colonies of the hermatypic coral species *Porites lobata* grow inter- and subtidally, covering only 2–7% of the benthos. The region is characterized by two seasons: a dry (December to May) and a rainy (June to November) season. The benthic community at Los Cóbanos is exposed to sedimentation impacts during the rainy season, when river runoff brings sediments and nutrients to the reef, significantly increasing water turbidity.

### Benthic community survey

The benthic community was evaluated using the Line-Point-Intercept-Transect method via snorkelling, as described by *English, Wilkinson & Baker (1997)*. In April 2018, seven 50 m long transects were placed parallel to the coastline at 2–3 m water depth, with at least 25 m between individual transects. During low tide, for a total of 100 points per transect (one every 0.5 m), the benthic organism underneath each point was identified in situ to genus level, except for turf and crustose coralline algae (CCA) that could not be identified to this level and were classified generically. If no organism was found on that point, non-living structures were categorized as: 'sand', 'rock' or '*Pocillopora* sp. skeleton'.
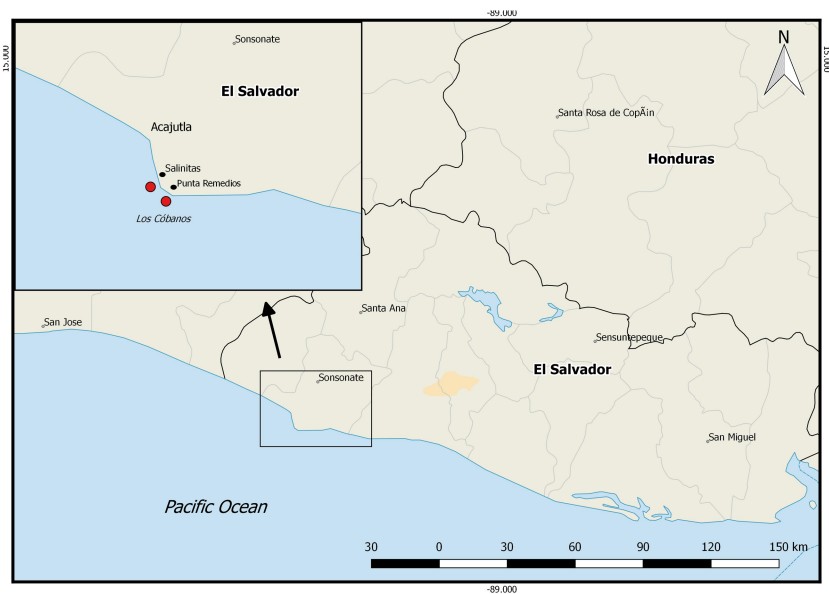

**Figure 1** **Pacific coast of El Salvador.** Square indicates Los Cóbanos reef where the monitoring and experimental set-up took place. Red dots: survey area. Map generated with QGIS2.18.13 (http://qgis.org).

The relative benthic cover of each category was calculated using the resulting 100 points per transect.

## Experimental cage set-up

The experimental design followed the Before-After Control-Impact/Treatment (BACI) design (*Stewart-Oaten, Murdoch & Parker, 1986*). Four $70 \times 70 \times 50$ cm$^3$ cages constructed from $2.5 \times 2.5$ cm$^2$ galvanized wire mesh (as used by *Smith, Smith & Hunter, 2001*) were deployed on the reef with at least 2 m distance from each other. Each cage enclosed one *Porites lobata* colony of a diameter of 8–12 cm and surrounding algae. Even though *P. lobata* only covered up to 4% of the benthos, it was targeted in the experiment to assess how simulated overfishing affects coral-algae interactions. Four more plots with a similar benthic community composition to the enclosed ones were selected as controls ($70 \times 70$ cm$^2$). The experiment ran for seven weeks, from April to May 2018. Each week the cages were cleaned using a plastic washing brush to avoid algae growth. The plots were inspected for small fish, snails, crustaceans and echinoderms every week during the cleaning. None were observed in the enclosed plots. Also, the control plots were carefully inspected weekly, but no macroinvertebrates were observed at any time. The cover of the different algae functional groups in the enclosed and control plots was estimated in situ using quadrats that indicated the cage limits ($70 \times 70$ cm$^2$) to the nearest 1% at the beginning and the end of the experiment. The same categories used during the benthic surveys were identified in the exclusion experiment. Only the uppermost benthic community layers were analyzed. The cover of *P. lobata* was estimated by comparing the projected area of the colony (calculated by measuring the length and width of the colony at the beginning and

end of the experiment) to the total selected cage and control area (0.49 m²). All surveys were conducted by the first author.

Due to logistical constraints, it was not possible to include semi-closed cages to assess potential physical effects of the cages in this experiment. CCA often perform better under lower light availability than other algae (*Van den Hoek et al., 1978*; *Vásquez-Elizondo & Enríquez, 2017*), while algal turfs increase their net primary productivity with increased water flow (*Carpenter, Hackney & Adey, 1991*; *Carpenter & Williams, 2007*). The cage structure could have reduced light availability and water flow within the enclosed plots, benefiting CCA. In order to quantify the potential effects of the cages on water flow and light attenuation, gypsum cards and HOBO light sensors were placed inside and outside cages with the same design as those used in the experiments, at the experimental sites for 24 h in April 2019. The gypsum cards were weighed before and after 24 h exposure to obtain an aggregate measure of water flow over that period. The HOBO light sensors measured light intensity within and outside the cages in lum/ft$^{-2}$ every 30 s.

### Data analysis

The mean relative cover of each functional group identified in the benthic community survey was calculated using the relative cover of all transects. To compare the community composition of the control and enclosed plots at the beginning and end of the experiment, Bray–Curtis Dissimilarity was calculated on untransformed data, and a PERMANOVA test was conducted using PRIMER 7. For the pair-wise comparisons, Monte Carlo $P$-values were obtained using PRIMER 7 due to the small number of permutations resulting from the PERMANOVA test, as suggested by *Anderson, Gorley & Clarke (2008)*. All additional tests were conducted in the statistics program R version 3.5.1 (*R Core Team, 2018*). Differences in coral, CCA, and turf benthic cover were tested with repeated measures ANOVA (rmANOVA from the ez package, *Lawrence, 2016*). In order to test the weight difference of the gypsum cards after the 24 h exposure, a $t$-test (from the stats package) was conducted. To test for potential differences in the measured light intensities, a Wilcoxon Rank Sum test (stats package) was used, as normality and sphericity assumptions were not met. An Anderson-Darling-Test (ADGofTest package) was used to test for data normality and Levene's-test (from the car package) was used to test sphericity assumptions. The jitter function (from ggplot2 package, *Wickham et al., 2019*) was used to add random variation to the non-metric multidimensional scaling (nMDS) plot and reveal the points with the same community assemblage that were overlapping.

## RESULTS

### Benthic community survey

The benthic community composition of Los Cóbanos reef was dominated by algae. Different types of algae comprised 72–98% of the benthic community. Turf algae were the most dominant algae group, with a mean benthic cover of 26.6 ± 8.8 (SD) %, followed by green algae with 22.8 ± 21.2 (SD) % (Table 1). Only one hermatypic coral species was found alive (*Porites lobata*), which had a benthic cover of 2.0 ± 2.8 (SD) %. Calcium carbonate skeletons of the branching coral genus *Pocillopora* were widely observed. Mobile benthic

**Table 1** Relative cover of the benthic community surveyed at Los Cóbanos reef using the Line-Point Intercept-Transect method.

|  | Genus/Group | Mean cover ± SD |
|---|---|---|
| Chlorophyta | *Codium* sp. | 7.9 ± 12.7 |
|  | *Halimeda* sp. | 14.9 ± 8.5 |
| Phaeophyta | *Padina* sp. | 5.4 ± 4.0 |
|  | *Ralfsia* sp. | 1.6 ± 1.9 |
|  | *Colpomenia* sp. | 0.3 ± 0.5 |
|  | *Dictyota* sp. | 8.6 ± 6.4 |
| Rhodophyta | *Galaxaura* sp. | 2.7 ± 2.7 |
|  | *Ceramium* sp. | 3.5 ± 2.9 |
|  | *Acanthophora* sp. | 3.3 ± 8.6 |
|  | CCA | 10.0 ± 3.1 |
|  | Rodoliths | 0.7 ± 0.8 |
| Turf | – | 26.6 ± 8.8 |
| Cnidaria | *Porites lobata* | 2.0 ± 2.8 |
| Rock | – | 2.0 ± 1.4 |
| Sand | – | 9.3 ± 8.5 |
| Skeleton | *Pocillopora* sp. | 7.5 ± 2.1 |

macroinvertebrates were not observed within the transects; however, the nudibranchs *Elysia diomedea* and *Glossodoris sedan*, and the echinoderms *Ophiocoma aethiops*, *Echinometra vanbrunti*, and *Holothuria (Halodeima) kefersteinii* were observed on the reef during the benthic surveys (Table S1).

## Fish herbivore exclusion

No significant effect of cage structures was observed on either light intensity (Wilcoxon Rank Sum test, $p = 0.520$) or water flow (*t*-test, $t_{(2.08)} = -1.21$, $p = 0.343$), indicating that the physical structure of the cages did not affect light or water flow.

There was a significant interaction effect of treatment and time on the benthic community composition (PERMANOVA, P(perms) = 0.001, perms = 996). The benthic community composition in the enclosed and control plots only differed significantly at the end of the experiment (PERMANOVA, P(perms) = 0.027, perms = 35, P(MC) = 0.007) (Fig. 2, Tables S2 and S3). The benthic community composition in the enclosed (PERMANOVA, P(perms) = 0.030, perms = 35, P(MC) = 0.001) and control (PERMANOVA, P(perms) = 0.028, perms = 35, P(MC) = 0.003) plots changed significantly between the beginning and the end of the experiment. Crustose coralline algae cover increased by a total 57.5% (from 0 to 57.5 ± 9.6 (SD) %) in the enclosed plots and 21.5% (from 1 ± 1.4 (SD) % to 22.5 ± 8.66 (SD) %) in the control plots, while the turf algae benthic cover decreased by 81% (from 83.8 ± 4.7 (SD) % to 2 ± 2.3 (SD) %) in the enclosed plots and 63% (from 79.75 ± 4.3 (SD) % to 16.5 ± 7 (SD) %) in the controls (Fig. 3). There was a significant interactive effect of treatment and time on both the CCA (rmANOVA, $F_{(1,6)} = 27.47$, $p = 0.003$) and turf algae benthic cover (rmANOVA, $F_{(1,6)} = 14.69$, $p = 0.009$).
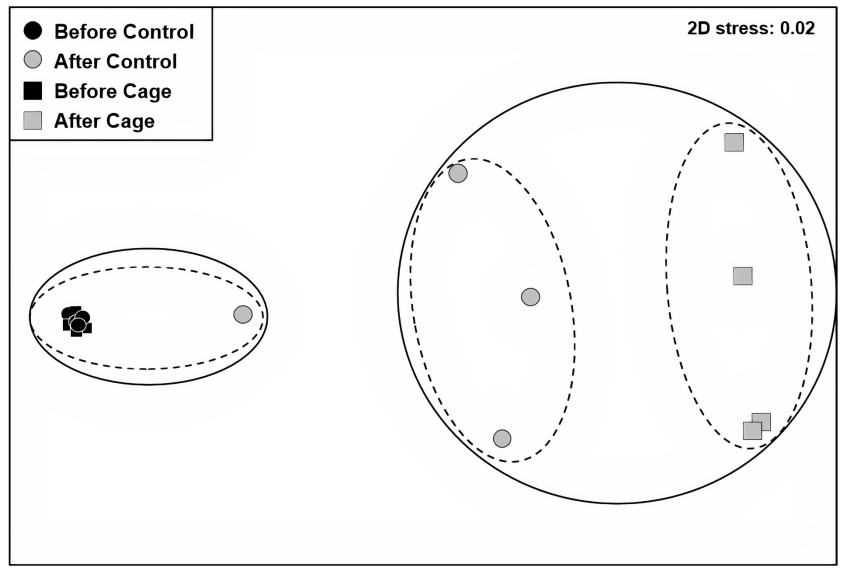

**Figure 2  Benthic communities before and after exlosure experiment.** Non-metric multidimensional scaling (nMDS) plot of control and enclosed (cage) benthic community composition before and after 7 weeks of experiment using Bray-Curtis similarity. Dashed circles: 60% similarity, continuous circles 40% similarity. Random variation added using the jitter function to reveal points overlapping (ggplot2 package, *Wickham et al., 2019*).

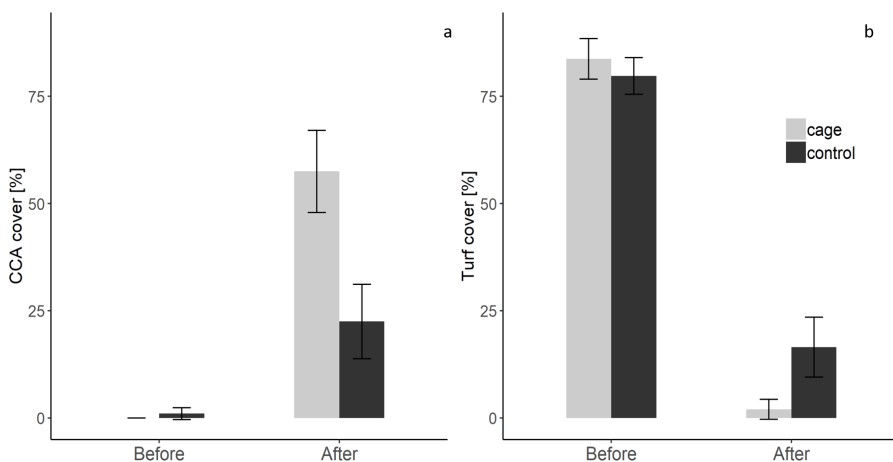

**Figure 3  Crustose coralline and turf algae cover in the enclosed and control plots before and after the experiment.** (a) CCA cover before and after the experiment (b) Turf algae cover before and after the experiment (Supplemental Information).

# DISCUSSION

## Benthic community survey

This study aimed to describe the current state of the benthic community of Los Cóbanos reef, El Salvador, and to investigate the effect of piscine macroherbivore exclusion on

benthic community composition. As expected (H1), the survey results show that the reef was dominated by algae (mostly turf and green algae) and that the coral cover was extremely low. Compared to the coral diversity described for the same reef by *Reyes-Bonilla & Barraza (2003)*, the number of living hermatypic coral species present decreased from five to one in less than two decades. Past unpublished surveys conducted at Los Cóbanos yielded similar results; however, the reported algae cover was slightly lower two years prior the present study (~72%) (*Herrera, 2017*). Yearly surveys conducted by J. Segovia using permanent transects reported a coral cover decrease from 6% to 2% between 2014 and 2016. *Segovia, Trejo & Ramos (2019)* and *Segovia (2020)*, using haphazardly deployed transects, documented a *P. lobata* cover of 2% in 2019 and 2020, in line with the value observed in this study for 2018. In contrast to the present study, the benthic community in Los Cóbanos has been mainly surveyed using 30-meter long transects at three different distances from the shore and a 1 m$^2$ quadrat (*Segovia, 2016*; *Segovia, Trejo & Ramos, 2019*; *Segovia, 2020*). Regardless of the difference in the monitoring methodology, the results of the different surveys yield very similar results, indicating a clear decrease of hard coral cover over time up to 2016, mainly attributed to El Niño events in 2014 and 2016 (*Segovia, Trejo & Ramos, 2019*). Moreover, according to personal communication with fishers from the local community, 20–30 years ago, *Pocillopora* sp. cover in the shallow parts of the reef was high. According to the fishers, the only way to access the reef was during high tide with the help of boats, suggesting a much higher coral cover and a higher hard coral diversity than the present one. A vast number of calcium carbonate skeletons, some of them still attached to the substrate, belonging to the coral genus *Pocillopora* were observed during the surveys, indicating this genus comprised a significant component of the shallow benthic community in the recent past.

Studies of other reefs in the ETP showed a higher hard coral species richness (2–3 coral species) than at Los Cóbanos (*Guzmán & Cortés, 1993*; *Alvarado et al., 2010*; *Stuhldreier et al., 2015*). Turf algae also seem to dominate other ETP reefs (*Cortés et al., 2017*; *Stuhldreier et al., 2015*). Yet, these reefs display a lower algae diversity, with only two or three macroalgae genera dominating the reef benthos (*Stuhldreier et al., 2015*), compared to nine different genera found at Los Cóbanos.

*Cortés et al. (2017)* state that the increasing temperature and length of El Niño events, sedimentation, and local human activities have caused reef degradation in the region. The coral cover of Los Cóbanos reef decreased drastically in the last decades. Two oil spills in El Salvador in 1993 together with extreme El Niño events may have caused the disappearance of most of the coral community of Los Cóbanos (*Cortés et al., 2017*; *Molina, 1996*; *Alvarado, 2012*; *Segovia, Trejo & Ramos, 2019*). Currently, the reef is exposed to high nutrient input from the local human population, intense fishing, and sedimentation (*Herrera, 2017*, unpublished; *Reyes-Bonilla & Barraza, 2003*). The hard coral population is limited to a small area at 0–15 m water depth extending for about 7,000 m along the coastline between Punta Remedios and Acajutla (Fig. 1) (pers. obs. J Segovia). The results of this study, together with the aforementioned literature, suggest that Los Cóbanos reef may have undergone a shift from a reef with high coral cover towards a rather algae-dominated reef, as described by *Pandolfi et al. (2005)* for other regions of the world, at least in the

first 600 m from the coast. *Bellwood et al. (2004)* defined coral reefs as "three-dimensional shallow-water structures dominated by scleractinian corals". The low coral cover and high algae cover and diversity underline that Los Cóbanos is no longer a coral reef but rather an algal-dominated reef. Concordantly, recent literature refers to Los Cóbanos reef as rather a rocky reef featuring hard coral and algal communities (*Herrera, 2017*, unpublished; *Segovia, 2017*). There is, however, no historical monitoring data that verifies the assumption that Los Cóbanos reef was indeed a coral reef in the past. Core drilling and examination of the underlying matrix might be able to resolve this question.

## Herbivore exclusion

Unexpectedly, CCA cover increased in both enclosed (by a total of ca. 58%) and control areas (ca. 21%), while turf algae decreased in both treatments (enclosed ca. 82%, control ca. 63%) at a remarkable speed. As CCA are among the slowest-growing marine algae, it is unlikely that this result reflects actual growth of CCA. Rather, at least some CCA may have been covered by turf-forming fouling epiphytes, the removal of which would have led to an apparent short-term increase in CCA. CCA are often considered as subordinate in their capacity to compete for space and are often overgrown or shaded by turf or macroalgae (*Dethier, 1994*; *Littler & Littler, 1980*). In some cases, this overgrowth even provides protection to CCA from harmful environmental conditions (*Figueiredo, Kain & Norton, 2000*). However, our results cannot confirm this, as only the uppermost layer of the benthic community was analyzed at the beginning and end of the experiment. No observations were made on whether CCA could indeed be found living under the turf algae. The exclusion of large piscine herbivores through caging had a significant effect on the benthic community composition. Surprisingly, there was a significantly higher apparent increase in CCA cover in the enclosed areas, whereas turf algae decreased more in the absence of piscine macroherbivores (Fig. 3). This is remarkable, as most literature indicates that herbivore exclusion causes an increase of macro- and turf algae (e.g., *Hughes et al., 2007*; *Roth et al., 2015*; *Thacker, Ginsburg & Paul, 2001*; *Zaneveld et al., 2016*). No effect of the cage structures on the current regime or light availability was detected through the 24 h measurements, meaning that the stronger observed benthic community changes in the enclosed plots were presumably caused by biotic factors. Nevertheless, the extent to which these results can be transferred to the seven-week experimental period is limited. High frequency data loggers were used to monitor light changes over a 24 h period on a day that was representative for the study period during the dry season. However, these measurements may not be representative for the study period which marked a transition from the dry to rainy season. Our study thus is not able to directly determine the biotic and abiotic processes behind the phenomenon observed at Los Cóbanos. Therefore, potential explanations for these counterintuitive results are discussed in the following paragraphs.

### *Low light and high nutrient availability*

The experiment at Los Cóbanos reef was conducted from April to the end of May, the transition months from the dry to the rainy season. During these months, there were several rainfalls which drastically decreased the underwater visibility through sediment input, and

possibly increased the nutrient concentration via river runoff. Low light availability has been reported to negatively affect turf algae growth (*Russell, 2007*). *Fricke et al. (2011)* attributed the decrease of turf biomass to the depth-related decrease in light quantity. The change in the light conditions could have reduced the turf algae, uncovering the CCA beneath it. However, little is known about the response of turf algal communities to changes in abiotic factors and further experiments would be needed to test this (*Fricke et al., 2014*). On the other side, CCA tolerate lower light availability (*Van den Hoek et al., 1978*; *Vásquez-Elizondo & Enríquez, 2017*) and have been reported to be nutrient-limited (*Smith, Smith & Hunter, 2001*). The sudden high nutrient input may have allowed CCA to survive the sudden change in light availability, contrary to the turf algae. Similar seasonal observations were made by *Menge, Lubchenco & Ashkenas (1986)* at a rocky shore in Panama, where the cover of coralline crust increased during the rainy season and decreased during the dry season. The change in abiotic conditions due to seasonal changes at the study site may have driven the unexpected turf algae decrease and apparent CCA increase in the enclosed and control plots.

*Exclusion of piscine macroherbivores*

In this study, the apparent CCA cover increase within the cages relative to the controls could have been caused by the exclusion of fauna that either feeds on turf algae or benefits its growth through, for example, farming behavior. Underwater visual fish censuses were conducted at Los Cóbanos during the study period; however, the results showed great variability. For this reason, the results were not used in this study. Nonetheless, the highly territorial turf algae farming fish species *Stegastes acapulcoensis* (*Dominici-Arosemena & Wolff, 2006*; *Robertson & Allen, 2015*) was found in high numbers in the census transects. As the presence of damselfishes can favor the growth of turf algae over CCA in algae dominated reefs (*Doropoulos et al., 2013*), the exclusion of *S. acapulcoensis* may have been to the detriment of turf algae, contributing to their strong decrease. A further explanation could be the exclusion of detritivores such as surgeonfish that remove sediments and detritus from turf algae (*Purcell & Bellwood, 1993*; *Tebbett, Goatley & Bellwood, 2017*). This could have benefited CCA in a similar way to observations by *Kendrick (1991)*, who carried out a study in the Galapagos archipelago and found that, after 51 days of experimentation, algal turf cover and recruitment decreased in treatments with a high rate of sedimentation favoring crustose coralline algae. This way, the exclusion of (a) key non-identified detritivore (s) could have magnified the effect of sediment deposition on turf algae, without detrimental effects on CCA.

*Grazing activity of macroinvertebrates*

A more probable explanation for the observed rapid decrease of turf algae and concomitant increase in CCA within the enclosed areas appears to be the grazing activity of smaller macroinvertebrates such as snails, sea urchins, or crabs. Following exclusion of larger fishes, invertebrate micrograzers have been shown to lead to shifts in community composition and reduced algal biomass (*Brawley & Adey, 1981*; *Zeller, 1988*). The deployed cages could have provided an accessible predator-free habitat for small benthic grazers. Rather than leading to a reduction in grazing, the exclusion of fishes may thus have increased

consumption of turf algae and other fouling epiphytes, uncovering previously hidden CCA. Although no macroinvertebrates were observed in or around the cages during the study, many macroinvertebrates are active at night, when the cages would have offered refuge from mesopredators. The sea urchin *Echinometra vanbrunti,* for example, was observed hidden between rocks during the benthic surveys in the reef. Furthermore, the weekly examinations were made in short periods during extreme low tides, when small invertebrates sought shelter from rising water temperatures and increased wave activity in crevices, tide pools or deeper parts of the reef. Additionally, no nocturnal observations were made that could verify the nocturnal grazing activity of sea urchins (*Nelson & Vance, 1979*; *Mills, Peyrot-Clausade & France Fontaine, 2000*). In exclusion experiments conducted on rocky shores in Galápagos and the Bay of Panama, when small grazers such as gastropods, crabs, and small fishes were not excluded, the CCA cover increased after 16 and 8 weeks, respectively (*Menge, Lubchenco & Ashkenas, 1986*; *Vinueza et al., 2006*). The CCA cover in the exclusion experiment on the Panamanian rocky shores was highest when only large fish were excluded from the areas (*Menge, Lubchenco & Ashkenas, 1986*), similar to the design in the present study.

This study's results suggest that biotic factors do affect the benthic community composition at Los Cóbanos. Even though CCA can suppress macroalgae in other ETP reefs (*Smith, Smith & Hunter, 2001*; *Vermeij, Dailer & Smith, 2011*), this experiment does not provide evidence that CCA could effectively outcompete turf algae, and that the observed increases were the result of actual CCA growth. If that were the case, the growth of CCA in this study would have, by far, exceeded the CCA growth rates reported everywhere else in the tropics (e.g., *Adey & Vassar, 1975*; *Villas Bôas, Figueiredo & Villaça, 2005*; *Tâmega & Figueiredo, 2019*). A more likely explanation for the apparent rapid increase of CCA cover is the loss of turf algae and other fouling epiphytes growing on CCA as a result of the grazing activity of small macroinvertebrates, thereby uncovering the CCA living underneath. CCA can survive overgrowth by filamentous turfs over long periods of time (*Kendrick, 1991*; *Airoldi, 2000*). *Lapointe (1997)* proposed that high nutrient availability and high grazing activity lead to CCA benthic dominance. Against expectations, simulated overfishing benefited calcifying algae (H2). Our results, however, highlight the potential importance of macroinvertebrates as grazers whose population seems to be controlled by piscine mesopredators at Los Cóbanos. Macroinvertebrates composed less than 1% of the benthic community in the reef. The underwater visual fish census also revealed a large number of piscine mesopredators at Los Cóbanos, such as the wrasses *Halichoeres dispilus*, *H. notospilus,* and *Thalassoma lucasanum* that feed on benthic invertebrates such as small crabs, snails and sea urchins (*Gomon, 1995*). The high number of mesopredators is possibly the result of overfishing of top predators such as sharks, barracudas, and large groupers, allowing mesopredators to proliferate (*Hixon, 2015*; *Prugh et al., 2009*). Future fishing management strategies at Los Cóbanos could focus on controlling the population of mesopredators by, for example, reducing the fishing intensity on top predators.

## CONCLUSIONS

It is most likely that the combination of grazing macroinvertebrates, increased nutrient concentration and turbidity as a result of seasonal river run-off, and potential effects of the cages, tipped the balance from turf algae to CCA. This phenomenon has been observed in other ETP reefs (*Menge, Lubchenco & Ashkenas, 1986*; *Vinueza et al., 2006*). However, due to the methodological limitations of the experiment, this study could not determine the drivers of the unexpected apparent increase in CCA cover observed at Los Cóbanos. Therefore, further experiments assessing the interaction between CCA and turf algae under different abiotic conditions at Los Cóbanos should be conducted. In addition, targeted studies of the fish and macroinvertebrate communities, their role in structuring the benthic community, and their trophodynamics are needed for a better understanding of the ecology of Los Cóbanos reef. Understanding the processes affecting the persistence of an important benthic component such as CCA is crucial first to understand the failed recovery of stony corals as in Los Cóbanos reef and secondly to take accurate management measures to avoid further deterioration.

## ACKNOWLEDGEMENTS

We thank to Ana María Velásquez and the rest of the park rangers for their support during this study's fieldwork. We also thank the reviewers for their insightful and helpful comments.

### Funding

The fieldwork of this study was funded by the Kellner und Stoll Stiftung, Bremen. Xochitl E Elías Ilosvay Master's studies were financed by the German Academic Exchange Service (DAAD). The funders had no role in study design, data collection and analysis, decision to publish, or preparation of the manuscript.

### Grant Disclosures

The following grant information was disclosed by the authors:
Kellner und Stoll Stiftung, Bremen.
German Academic Exchange Service (DAAD).

### Competing Interests

The authors declare there are no competing interests.

### Author Contributions

- Xochitl E. Elías Ilosvay conceived and designed the experiments, performed the experiments, analyzed the data, prepared figures and/or tables, authored or reviewed drafts of the paper, and approved the final draft.
- Johanna Segovia and Walter Ernesto Elias performed the experiments, authored or reviewed drafts of the paper, and approved the final draft.

- Sebastian Ferse and Christian Wild conceived and designed the experiments, authored or reviewed drafts of the paper, and approved the final draft.

## Field Study Permissions

The following information was supplied relating to field study approvals (i.e., approving body and any reference numbers):

The Ministry of Environment and Natural Resources of El Salvador (MARN, initials in Spanish) approved the fieldwork for this study inside this nature reserve: ''Complejo Los Cóbanos''.

## Data Availability

Raw data are available in the Supplemental Files.

## Supplemental Information

Supplemental information for this article can be found online at http://dx.doi.org/10.7717/peerj.10696#supplemental-information.

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
