# Peer review of "Rapid relative increase of crustose coralline algae following herbivore exclusion in a reef of El Salvador"

_PeerJ, doi:10.7717/peerj.10696_

## Round 0.1 · original submission · Major Revisions

Three expert reviewers have evaluated your manuscript and their comments can be seen below. As you can see these range from accept to major revisions. Overall, this submission is interesting and relevant. However, there are a number of issues that have been brought up that mostly relate to providing more details in the methods and experimental design and its potential limitations, as well as a reinterpretation of the results as suggested by two of the reviewers.

Reviewer 1 ·

Basic reporting

Elías Ilosvay and colleagues perform a fish exclusion experiment and survey benthic cover. The methods are appropriate, and the results are clear. The responses to reviewers are complete. I really see no problems. I recommend accepting the manuscript.

Experimental design

Appropriate

Validity of the findings

Strong

Reviewer 2 ·

Basic reporting

no comment

Experimental design

The replication may be an issue and more detail on replication and how that relates to figure 2 is needed.

Validity of the findings

A concern would be the potential for another factor leading to the increase in coralline cover which would be the use of a different data recorder or surveyor for the later dataset leading to differences in cover due to human sampling (over-estimation by person sampling later) and not ecology.

Additional comments

I believe this manuscript to be within the scope of PeerJ. This study was generally well designed, depending on the number of “before” samples collected the study may be well replicated, and was conducted for 7 weeks to determine responses to experimental treatments. The overarching goal was to determine how fishing may affect Los Cóbanos reef. Researchers addressed this by investigating changes in the benthic community following the exclusion of piscine macroherbivores using cages over a period of seven weeks. The cages may have provided refuge for herbivorous invertebrates which graze fleshy macroalgae and complicate interpretation of the results. The paper generally reads well, is interesting, and should be of interest to both the research and applied ecology communities. My main concern would revolve around the interpretation of the results (similarly to the previous reviewer comments). My main concern would be the potential for another factor leading to the increase in coralline cover which would be the use of a different data recorder or surveyor for the later dataset leading to differences in cover due to human sampling (over-estimation by person sampling later) and not ecology. The number of replicates used for the caging experiment may be a concern. I did not see any before control points in figure 2? I was expecting to see four control points and four points where cages will be installed (randomly selected) before the experiment and then four control points and four cage points after 7 weeks. Im a little concerned about replication or maybe I missed something. If these concerns could be alleviated then there are only minor comments.

General comments.

1) Line . 48-50 - Could state how fishing indirectly leads to the favor of “competitive fleshy algae and turf”. For example, “By removing herbivorous fishes that consume macroalgae, fishing leads to release of competitive fleshy …”. The connection between fishing and algae/turf seems direct as previously written.
2) Line 152-155. Im unclear on what “resist” means. “CCA tends to resist lower light…”. I think generally CCAs perform well in lower light levels but I think youre making the case CCAs may be less common in lower light as they ”resist” lower light. Maybe you meant they can tolerate lower light or perform better under lower light? It looks like the next sentence supports CCAs may do well in low light so id clarify.
3) Line 159-164 – These results for water flow and light differences within and outside of cages could be reported in the results section rather than the methods section. These results help with the interpretation of the caging effects.
4) Line 163-164 – Id maybe reword this. “any” unwanted physical cage effects could include many other factors (alteration of sedimentation, structure stops drifting algae, structure attracts mobile fauna, etc.). Id consider rewording to “indicating that the physical structure of cages do not affect light or water flow“ and not “any unwanted physical effects”.
5) Line 179 – I believe the test is Levene’s Test not Levene Test.
6) Line 198 – I believe that with such low numbers of permutations (perms = 35) its recommended to conduct monte carlo simulations.
PERMANOVA+ for PRIMER: Guide to Software and Statistical Methods by Marti J. Anderson, Ray N. Gorley & K. Robert Clarke (2008) - “The last column of the PERMANOVA table (‘Unique perms’) indicates how many unique values of the test statistic were obtained under permutation. Recall that PERMANOVA does not systematically do all permutations, but rather draws a random subset of them. In the example (Fig. 1.10), this value is very large (9862) and close to the number of random permutations that were chosen to be done by the user (9999). This means that only a few repeated values of pseudo-F? were encountered under permutation, and the number of unique values is plenty enough to make reasonable inferences using the resulting permutation P-value, as shown. This information is important because, in some cases, the number of possible permutations is not large, and very few unique values of the test statistic are obtained. In such cases, a more meaningful (but approximate) P-value can be obtained by random sampling from the asymptotic permutation distribution instead (see the section Monte Carlo P-values below).”
7) Line 204 – add “and” before “turf algae benthic cover (…”.
8) Line 194 – 205 – I would think a concern may be that the conclusions could be due to factors other than dramatic increases in CCA cover that have been brought up before by reviewers. One factor that could yield the results would be different people collecting the data. Maybe I missed where this was stated but I would like to know if the same surveyor took the data as this is commonly the first thing of concern when taking community data. Are changes due to communities responding to time, space, or experimental treatment or are differences due to how the data were collected.
9) Line 265 – No need for “benthic” cover. CCAs are benthic and just CCA cover would work.
10) Line 186 – I think this should be high “nutrient” availability without the s.
11) Line 295 – CCA tend to “resist” lower light. As stated above I think CCAs perform well at low light and may even perform optimally depending on which species and whether they are in polar versus tropical areas. I think there may be a better term than resist. If youre stating they perform better at lower light relative to turf algae maybe use “tolerate lower light”.
12) Line 316 – After 51 days of “the” experiment or after 51 days of “experimentation”.
13) Line 340-341 – I don’t quite get this statement. Id reword “…the CCA cover increased after 16 and 8 weeks already, respectively…”. Maybe …”After 8 and 16 weeks, respectively”?
14) Line 354 – Maybe reword ”CCA have been found to survive overgrow by filamentous turfs over long periods of time” to “being overgrown” or “overgrowth”.
15) Figure 2 – I don’t see any before control (black circle) points on the figure? I only see 2 points for the before cage (black square)? Was there only two before surveys for plots? I think I may have missed the replication. Line 138-139 – “Plots with a similar benthic community composition to the enclosed ones were selected as controls (70 x 70 cm2)”. How many plots? I don’t see a clear description of replicates for control/treatment for before and after. Figure 2 states "dotted lines" but I think that should be "dashed lines".
16) Figure 4b – caption could be reworded. Relative benthic cover of turf algae in enclosed and control areas, before and after the experiment. Figure 4a should have a similar caption to 4b but lacks the statement “before and after the experiment”.

Reviewer 3 ·

Basic reporting

Ilosvay et al. survey an impacted coral reef off the coast of El Salvador and use experimental grazer exclusion to explore the effects of herbivorous fish and exclude these species to simulate over-fishing. The most interesting part of the study (in my opinion) is the reaction of coralline algae and turf algal species to these cages. The authors touch on this in the discussion, but I feel this is a story about turf reduction and not necessarily an increase in corallines.

Experimental design

Can you expand on the methods you used to measure percent cover before the cages were installed? Were different layers not analyzed? This topic is touched upon in the discussion, but possibly these results show a decrease in fleshy seaweed and not an increase in corallines.

While I understand the logistical limitations of open cages, I'm not sure a 24 h period measuring flow and light is enough to show that flow and light did not affect your results. Do you feel the 24 h period measured was representative of the study period?

It seems that there were grazers around, but that your sampling methods did not catch them, although I think that you discussed this, I think non-fish grazers might be a big part of these results.

My other main concern is turf numbers dropped in the control plots as well, and I think this indicates an abiotic factor playing a role and then the biology magnifying this. I think in your discussion you need to touch on multiple factors affecting your results. Such as season and cage effects.

Validity of the findings

The findings of this study are valid and important. But I do feel that potentially season and the cages played a role in the results. In the discussion the topic of multiple stressors needs to be discussed. The limitations of the experimental design need to be explained more as well. Even with these limitations, I do think there are interesting patterns that emerged from this study.

Additional comments

See above text.

---

## Round 0.2 · Minor Revisions

Two expert reviewers have evaluated your manuscript. Both agree that the manuscript is acceptable for publication in PeerJ. However, one of the reviewers has suggested a couple of minor edits that you should consider.

Reviewer 2 ·

Basic reporting

no comment

Experimental design

no comment

Validity of the findings

no comment

Additional comments

I think the authors did a great job at addressing the reviewer concerns and I am supportive of this manuscript being published. Some small comments were to potentially add a clarifying statement to line 202-204 that the jitter function was used to reveal points with the same community assemblage that were overlapping and line 331 changing "...period from dry to rainy season" to "...period from the dry to rainy season". The sentence from 437-440 was quite long. One option may be "It is most likely that the combination of grazing macroinvertebrates, increased nutrient concentration and turbidity as a result of seasonal river run-off, and potential effects of the cages, tipped the balance from turf algae to CCA".

Reviewer 3 ·

Basic reporting

Pass

Experimental design

Pass

Validity of the findings

Pass

Additional comments

Thank you for your consideration of my comments. Great work!

---

## Round 0.3 · Minor Revisions

Your response to point 1 in your rebuttal letter does not coincide with the text in the tracked changes or the final version of the manuscript: "to the non-metric multidimensional scaling (nMDS) plot" is in the rebuttal letter but not the manuscript. Is this an oversight?

---

## Round 0.4 · accepted · Accept

I am satisfied with the changes made to the manuscript.